

# Posture similarity index: a method to compare hand postures in synergy space

Nayan Bhatt and Varadhan SKM

Department of Applied Mechanics, Indian Institute of Technology Madras, Chennai, India

## ABSTRACT

**Background**. The human hand can perform a range of manipulation tasks, from holding a pen to holding a hammer. The central nervous system (CNS) uses different strategies in different manipulation tasks based on task requirements. Attempts to compare postures of the hand have been made for use in robotics and animation industries. In this study, we developed an index called the posture similarity index to quantify the similarity between two human hand postures.

**Methods**. Twelve right-handed volunteers performed 70 postures, and lifted and held 30 objects (total of 100 different postures, each performed five times). A 16-sensor electromagnetic tracking system captured the kinematics of individual finger phalanges (segments). We modeled the hand as a 21-DoF system and computed the corresponding joint angles. We used principal component analysis to extract kinematic synergies from this 21-DoF data. We developed a posture similarity index (PSI), that represents the similarity between posture in the synergy (Principal component) space. First, we tested the performance of this index using a synthetic dataset. After confirming that it performs well with the synthetic dataset, we used it to analyze the experimental data. Further, we used PSI to identify postures that are "representative" in the sense that they have a greater overlap (in synergy space) with a large number of postures.

**Results**. Our results confirmed that PSI is a relatively accurate index of similarity in synergy space both with synthetic data and real experimental data. Also, more special postures than common postures were found among "representative" postures.

**Conclusion**. We developed an index for comparing posture similarity in synergy space and demonstrated its utility by using synthetic dataset and experimental dataset. Besides, we found that "special" postures are actually "special" in the sense that there are more of them in the "representative" postures as identified by our posture similarity index.

Corresponding author
Varadhan SKM, skm@iitm.ac.in

## INTRODUCTION

Accurate modeling of hand and finger postures is of interest in robotics and animation industry. In the animation industry, differentiation of one hand posture from another is crucial for the perception of a gesture and to bring in a more "lifelike" appearance (*Hoyet et al., 2012*). In robotics, the comparison of a robotic posture with the biological (human) hand posture is useful to assess the performance of a robot (*Feix et al., 2013*).

Studies in perception of animations have exploited the relatively low number of dimensions that are required to distinguish between perceived postures (*Hoyet et al., 2012*). They used a relatively small number of markers to produce postures that are perceived as different. Studies in robotics have focused on the development of methods to compare a robotic posture with hand posture, the "anthropomorphism index" (*Feix et al., 2013*; *Romero et al., 2010*). Other works have focused on exploring the low-dimensional subspace used by human hand postures (*Ciocarlie & Allen, 2009*; *Wheatland, Jörg & Zordan, 2013*). These studies are performed in line with the hope that understanding human grasping could improve the design of grasping robots. These approaches attempt to compare human hand postures with those generated by a computer (animations) or by a robot (robotics). These studies have not attempted a comparison of a human hand posture with another human hand posture.

In the field of motor control, several studies have attempted to characterize and model hand postures (*Santello, Flanders & Soechting, 1998*; *Gentner & Classen, 2006*; *Cavallo et al., 2016*; *Leo et al., 2016*). Many of these studies have used Principal Component Analysis (PCA) to identify the dominant dimensions in the multi-dimensional space of a set of hand and finger postures (where a set of joint angles is used to quantify each posture). Many of these studies quantified and presented the proportion of variance explained by these dominant dimensions.

However, these studies have not attempted comparison of postures per se. As such, assessment of similarity is a challenging problem. Some relatively sophisticated algorithms have been proposed (e.g., *Roweis & Saul, 2000*) to assess similarity. We believe that a method to specifically compare human hand postures in synergy space would be useful in the field of motor control, animation, and robotics. We attempt to develop such a method in this paper.

In this study, we propose, define, and develop an index (posture similarity index, PSI), that quantifies similarity in the principal component ("synergy") space. We first test this index on a synthetic dataset and then with a small experimental dataset. We then use this index to study similarity across a large number of postures (100 postures, including both everyday life and special postures). Based on PSI, we identify representative postures that may reflect dimensionality of hand action in the synergy space. These representative postures are those that have a relatively large overlap in synergy space with several postures. Among this set of "representative postures", we further compare the number of everyday life postures and "special" postures.

## MATERIALS & METHODS

### Participants

Twelve right-handed volunteers (six males and six females (Age: 25.5 ± 1.55 (SD) years) provided written informed consent and participated in the study. No participant had a history of any neurological disorder or any trauma to the upper extremity. All experimental procedures were approved by the Institutional Ethics Committee of IIT Madras (Approval number: IEC/2016/02/VSK-1/11). All participants were naive to the purpose of the study.

## Experimental procedure

The experimental procedure involved performing simple motor acts including specific hand postures and object manipulation. For the collection of kinematic information, we used electromagnetic tracking sensors (diameter 1.8 mm, resolution 1.27 μm, static position accuracy 0.76 mm, static angular orientation accuracy 0.15 degree; model: Liberty micro sensors; Polhemus Inc., Colchester, VT, USA). For preventing interference, we minimized the number of metallic objects in the experimental room. We placed the sensors on the dorsal side of digits (I: index; M: middle; R: ring; L: little) using double sided adhesive tape as shown in Fig. 1A. These sensors captured the movement of distal segment/phalanx, middle segment/phalanx, proximal segment/phalanx, and metacarpals. From these quantities, we derived the distal interphalangeal (DIP), proximal interphalangeal (PIP), and metacarpophalangeal (MCPs) joint angles for I, M, R, and L. For the thumb, we measured the movements of distal, proximal, and metacarpal segments and we computed the interphalangeal (IP), metacarpophalangeal (MCP), and carpometacarpal (CMC) joint angles. We mounted the reference sensor near proximal part of the wrist between 2nd and 3rd metacarpal bone. We designed a customized user interface in LabVIEW (National Instruments) environment to collect data at 120 Hz, and performed offline analysis using MATLAB (MathWorks, Natick, MA, USA).

## Kinematic data acquisition

The experimental setup is presented in Fig. 2. Placement of sensors in the lab coordinate system was designed according to biomechanical standards (Wu et al., 2005). We defined a base posture at the start of every trial. In this base posture, the hand is in pronation position with all the fingers (including the thumb) fully extended and adducted. Participants were seated in a height adjustable chair comfortably with his/her arm resting on the table at approximately wrist level, with palm facing the surface of the table. A 24-inch LED monitor was placed approximately 1 meter away from the participants.

The experiment consisted of two types of tasks. The first type of task called the externally constrained postures included 30 postures that involved manipulation of an object. The second type of task called the internally constrained postures, included 70 free hand postures that did not involve object manipulation, hence constrained only by the CNS. The participants viewed pictures of target postures for internally constrained postures (i.e., American Sign Language (ASL), Indian dance form Bharatanatyam) on a computer screen. At the beginning of each trial, the participant started from the base posture and reached the target posture. Each trial was five seconds long, and was repeated five times. Hence, participants performed a total of 500 (100 postures × 5 trials per posture) trials. There was a rest period of ten seconds between trials. Additional rest was provided during the experiment whenever the participant requested.

For the externally constrained postures, before each trial, an object was placed on a predefined marked position, nearly 15 cm away from the hand. Participants were then asked to lift and hold the object approximately 10 cm above the surface. The experimenter visually compared the static posture at the end of the trial with the target posture. The experimenter also carefully observed all the trials and repeated any trial in which the posture

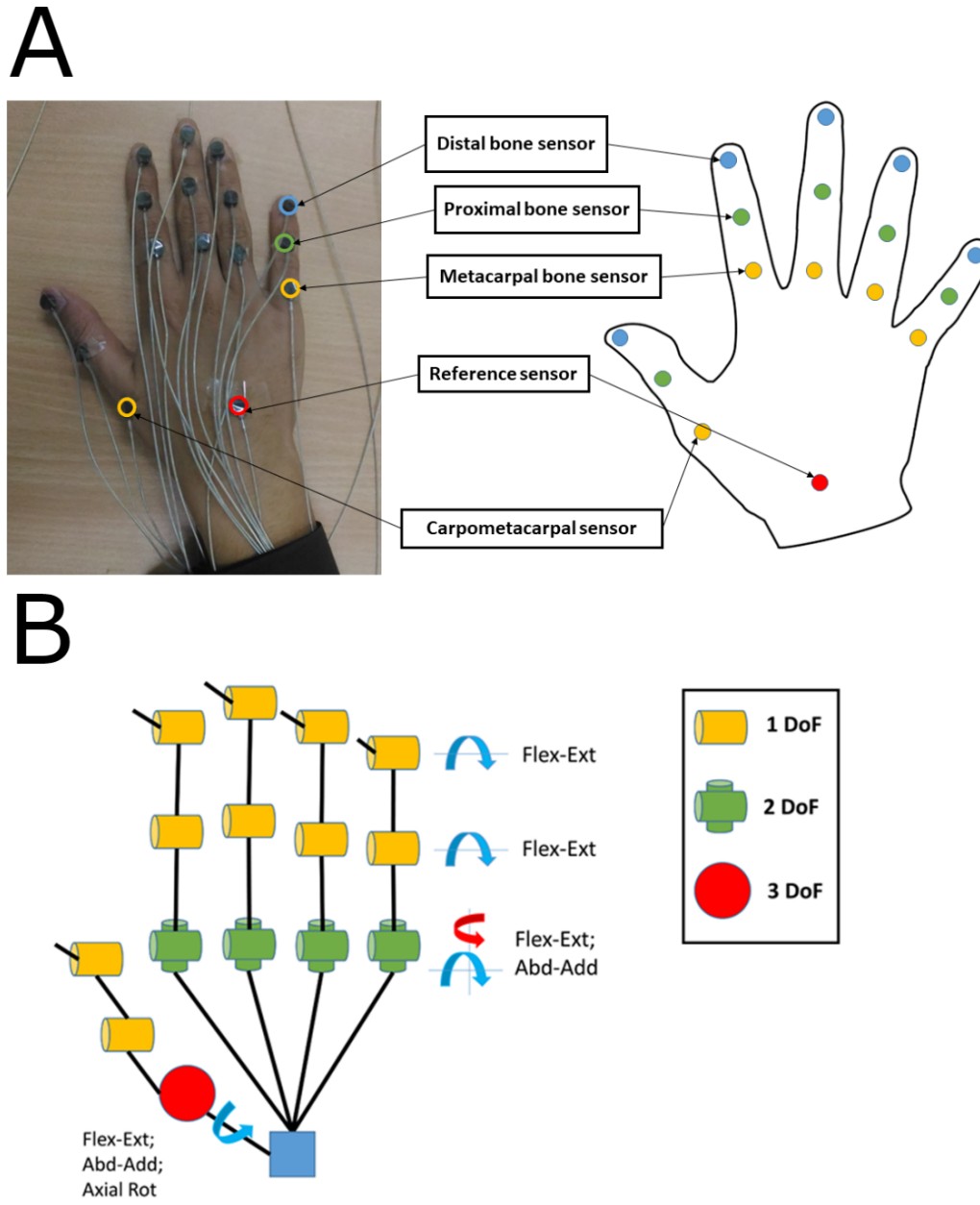

**Figure 1** **(A) Sensor placement location on hand. (B) Kinematic hand model consisting of 21 DoFs.** (A) Fifteen sensors are placed on digits and one reference sensor is placed near wrist. (B) DIP and PIP joints corresponds to one DOF; MCP joint corresponds to 2 DOFs; for thumb CMC joint corresponds to three DOFs. All photographs by Varadhan SKM. Hand model: Nayan Bhatt.

was not faithfully performed. On an average, less than 2% of the trials were repeated across participants. We have presented the set of all the postures in Supplemental Information 2.

## Data processing and kinematic model

We computed all joint angles using the principles of mechanics (*Zatsiorsky, 1998*; *Jazar, 2010*). We considered the trial duration from ∼3.6 to 4.25 s as static posture (i.e., 70

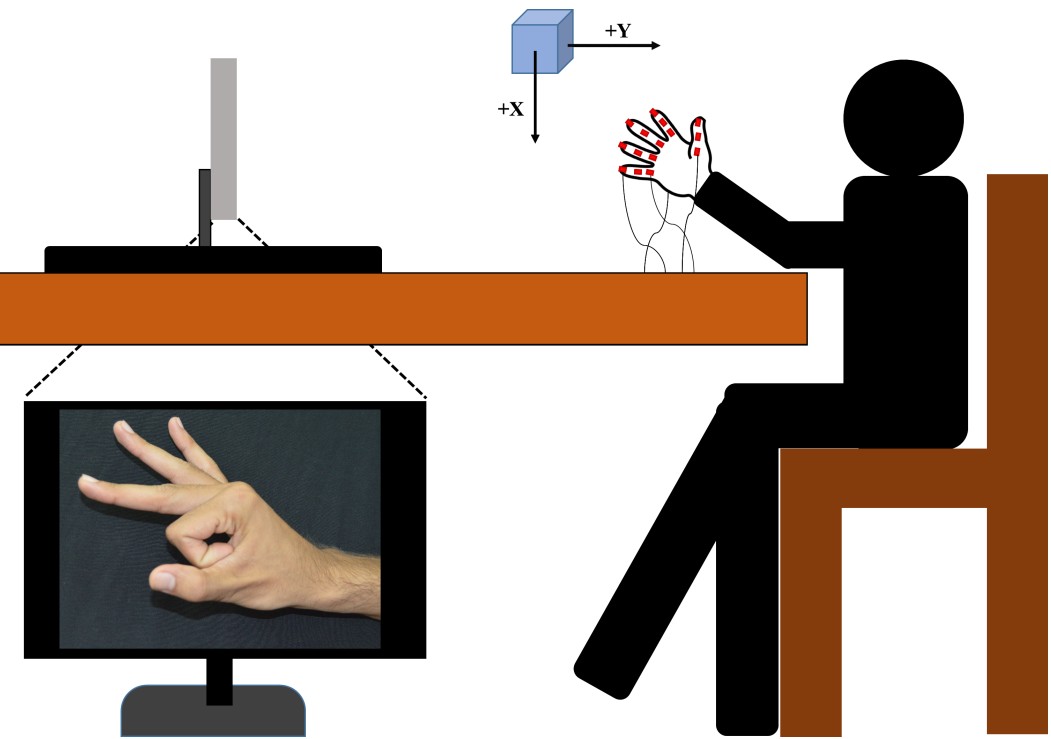

**Figure 2  Experimental setup.** Participants were seated in a height-adjustable chair. Placement of the transmitter was based on biomechanical standards (*Wu et al., 2005*). The computer screen was placed approximately 1 m away from the participant. Image of posture was shown to the participant on the screen. Participants were asked to perform posture which is shown on the screen starting from the base posture.

samples) because the change in joint angles were not significant after 3.6 s across postures and we dropped the last few samples to avoid the end of the trial effects. Joint angle data was filtered using second-ordered zero-lag low pass Butterworth filter. For removing any possible effect of physiological tremors in static postures, we set the cutoff frequency at 4.5 Hz. The frequency content of these tremors has been reported to be in multiple ranges of frequencies (*Deuschl et al., 2001*; *Fahn, Jankovic & Hallett, 2011*; *McAuley & Marsden, 2000*). The general agreement between these studies is that tremor frequencies are below 5 Hz, although maximum frequencies vary. Our goal was to compare static postures (not movements); hence we chose a low value of cutoff frequency. We constructed a matrix of size 35,000 (5 trials × 100 postures × 70 samples per posture) × 21 (dimension) from the entire data for each participant, and used this matrix for further analysis.

The hand model had 21 degrees of freedom (DoF) as shown in Fig. 1B (similar to *Rezzoug & Gorce, 2008*). In this model, each finger was considered to have four DoFs: it included flexion-extension (Flex-Ext) of DIP and PIP joints; it also included flexion-extension and abduction-adduction of MCP joint. We represented the thumb with five DoFs which included flexion-extension of IP and MCP joint, and three DoFs which included flexion-extension, abduction–adduction, and axial rotation from CMC joint.

## Exploring kinematic synergy space using a linear method

There is a strong relationship between DIP and PIP joint angles for I, M, R, and L. DIP Vs PIP joints across twelve participants showed a linear relationship ($r^2 = 0.997$) between these joints across fingers. Dependency and amount of co-dependency may vary. However, covariation patterns are stereotypical. For quantification of variance explained by PCs, we used Principal Component Analysis (PCA). Static hand postures were transformed from joint angle space to space with basis vectors orthogonal to each other (i.e., "synergy space") using the PCA method. The dataset (35,000 samples = 5 trials × 100 postures × 70 samples per posture) were analyzed. Dimensionality of the dataset is 21 considering 21 DoFs hand model. We performed PCA by centering the data and using Singular Value Decomposition (SVD) algorithm.

## Posture similarity index (PSI)

Comparing two different hand postures in the synergy space, rather than in joint angle space, may provide insightful information about overlap (in synergy components) between two postures. The measure developed by us compares two hand postures in synergy space and gives a quantification number called Posture Similarity Index (PSI).

## Theoretical formulation

Joint angle space spanned by various hand movements is also termed as *action manifold* (*Feix et al., 2013*). Our objective is to find a subset of postures that may help in exploring 21-dimensional space in the best possible way. We also included a set of extreme postures in the study to explore individual degrees of freedom of hand. One approach to study hand postures in synergy subspace is dimensionality reduction (*Santello, Flanders & Soechting, 1998*). An alternative view is that the control signal helps controlling multiple degrees of freedom simultaneously (*Todorov & Jordan, 2002*; *Todorov & Ghahramani, 2004*). In synergy space, some postures might be similar to others. Due to such similarity between postures, it is possible to derive one posture by minor modification in synergies of a slightly different posture. However, if two postures are very different, it is not expected that minor modification of synergy of one posture will lead to the other posture. Quantifying the similarity between postures will help us find the representative postures. For each participant, the dataset contained 100 static hand postures and 35,000 (5 trials × 100 postures × 70 samples per posture) samples with 21 dimensions. Since there is no learning or history effect between trials of a posture, we selected a random trial from five trials for every posture separately, resulting in 100 selected trials per participant (one trial per posture). We assumed that any one of the five trials is representative of the 5 trials performed. Note, however, that the "average" of the five trials may not represent each individual trial. We used PCA to project static hand postures from joint angle space to synergy space. Before applying PCA, we performed column-wise normalization for all joint angles.

Let the original dimensions of the data space be D (in this case dimensionality is 21, refer to Fig. 1B). The projection matrix (W, dimension N × N) was created using all static posture samples (35,000) that transformed static posture from joint angle space

to multidimensional synergy space. All static postures were projected along eigenvectors separately using projection matrix (W) giving projected values (70 × N). Note that in our case, static posture has dimensions of 70 × 21 (70 data points and 21 joint angles).

The covariance matrices ($\Sigma(i)$, dimension N × N (i.e., 21 × 21); for $i = 1$ to 100) were computed from projected postures in feature space for each posture (70 × N) separately. Diagonal elements of $\Sigma$ represent the amount of variance in each PC direction. Variances ($\sigma^2$) are expected to be similar for similar postures. The ratio of variances (i.e., $F = \sigma_{1,1}^2/\sigma_{1,2}^2$) for two similar postures will be closer to 1. If postures are not same, the weighted ratio (i.e., $\lambda F$) will give two different values. To avoid mathematical ambiguity, we always chose $F$ less than one in all cases.

Based on eigenvalues ($\lambda$) of PCs, weights were assigned to $F$ ratio. For generating *posture similarity index (PSI)*, we performed summation across all the $N$ PCs. *PSI* is expected to be higher for two similar postures in synergy subspace.

$$F_n^{i,j} = \frac{\sigma_{ni}^2}{\sigma_{nj}^2} \tag{1}$$

where $F$ is the ratio of two variances in the $n$th PC direction.

$$PSI(i,j) = \sum_{n=1}^{N} \lambda_n F_n^{i,j} \tag{2}$$

where $i, j$ represent similarity between the $i$th and $j$th posture. $\lambda_n$ represents Eigen value of $n$th Eigen vector. PSI is the weighted sum of all variance ratios across $N$ dimensions.

### Evaluation of PSI on synthetic data

To test the performance of our approach, we decided to first test the algorithm on a synthetic dataset. Three synthetic postures were generated using multivariate normal distribution using the following mean values and standard deviations. For posture 1 (mean ± S.D in $X, Y$, and $Z$, 5 ± 1.85, 2 ± 1.78, 10 ± 2.42), for posture 2 (mean ± S.D in $X, Y$, and $Z$, 10.54 ± 1.85, 7.35 ± 1.78, 17.25 ± 2.42). Mean for the second posture is generated using the following equation:

$$\text{Mean}(p2) = \text{mean } p1 + 3 * \sigma(p1). \tag{3}$$

For posture 3, we used mean ± S.D in $X, Y$, and $Z$, 5.69 ± 0.694, 2.611 ± 0.61, 11.021 ± 1.02 for generating data points. Mean for the third posture was near to first posture since we wanted to overlap the data. Mean for the third posture is calculated using the following equation:

$$\text{Mean}(p3) = \text{mean } p1 + 1 * \sigma(p3). \tag{4}$$

Using these, we generated synthetic postures, each containing 500 samples. The variance of posture 1 and posture 2 are same which is (3.42, 3.17, 5.83), the variance of posture 3 is smaller than other postures (0.48, 0.37, 1.043). Mean of posture 1 (represented in red) and posture 3 (represented in blue) are closer to each other and mean of posture 2 (represented

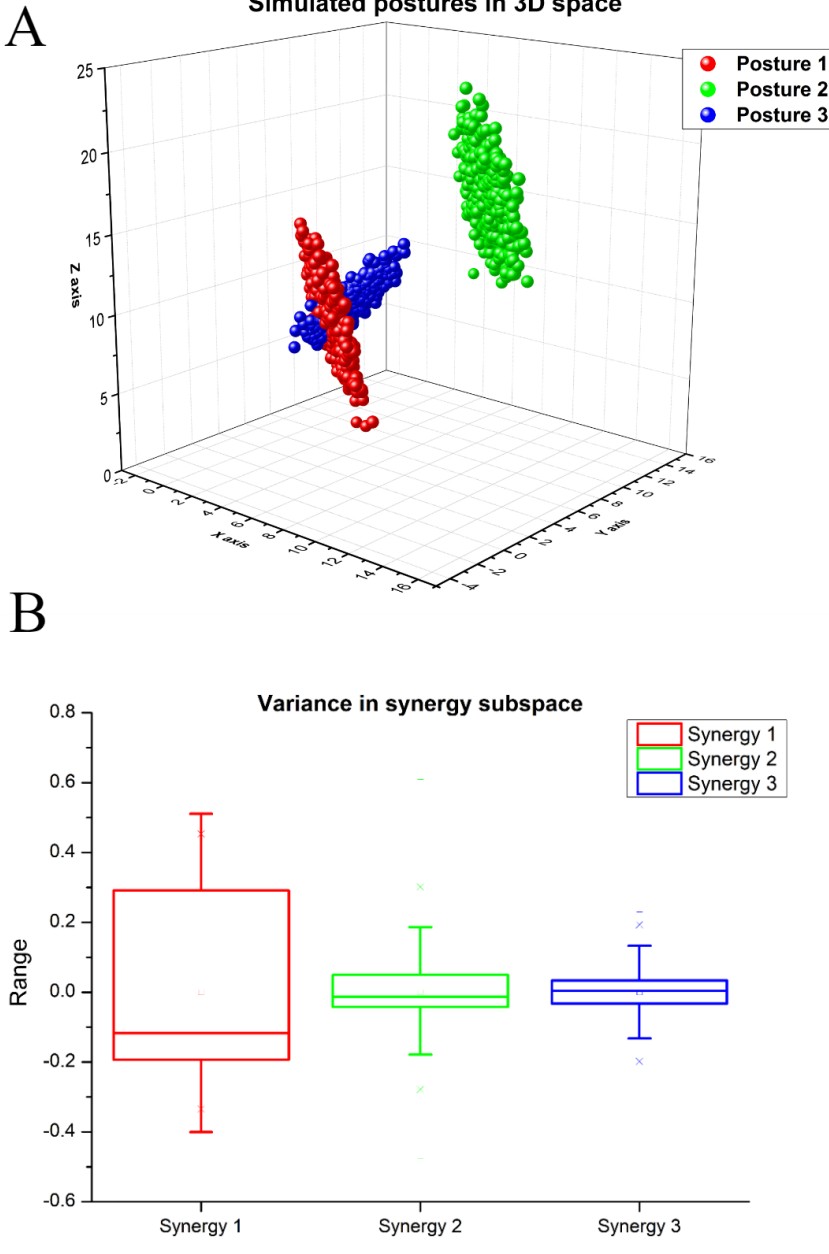

**Figure 3** **(A) Simulated postures in 3D space. (B) Variance in the synergy space for synthetic dataset**
(A) Posture 1 and posture 2 have same covariance matrix but different mean values. Mean of posture 3 is
closer to posture 1, which means there is an overlap between posture 1 and 3. The structure of posture 1
and 3 are different. (B) Synergy 1 represents highest variance (i.e., 77.8%), Synergy 2 explains ( 15.8%)
variance, and Synergy 3 explains ( 6.7%) variance.

in green) is far. Variance of posture 1 and 2 are similar but posture 3 has smaller variance,
and orientation of posture 3 data is different in the space. There is an overlap between
postures 1 and 3 as shown in Fig. 3A. We deliberately chose such a dataset to check if our
method could capture the details described above.

We normalized the data before the application of PCA. For extracting kinematic synergies from the simulated dataset, we performed PCA. The 1st principal component (synergy) explains approximately 77% of the variance, the 2nd principal component explains approximately 15% of the variance, and the 3rd component explains approximately 6% of total variance, as illustrated in Fig. 3B.

We used projection matrix W ($3 \times 3$) to project data ($1{,}500 \times 3$) on synergy space/global PC space. We computed covariance matrices ($\Sigma_i$ $3 \times 3$, for $i = 1{:}3$) for the projected dataset. Diagonal elements of the covariance matrix represent the projection of each posture in PC direction.

The projection ratio in each PC direction is computed and presented in Table S1.

Each ratio ($F_n^{i,j}$), computed as given in Eq. (1) is multiplied with respective Eigen values ($\lambda$s) and finally added. We performed this exercise for all the postures and developed the posture similarity matrix. Each element in the matrix represents *posture similarity index* comparing the amount of similarity between the two postures.

The ratio is selected based on the condition that we chose $F$ to be less than unity. The magnitude of each Eigen vector is 0.062, 0.0127, and 0.0054. Elements for the *PSI* matrix are computed using Eq. (5).

$$PSI_{i,j} = \sum_{n=1}^{3} \lambda_n * F_n^{i,j}. \tag{5}$$

From the matrix of posture similarity in Table S2, we can observe that postures 1 and 2 are more similar in comparison to postures 1 and 3 and postures 2 and 3. Postures 1 and 2 have the same variance and structure with the mean shifted. However, posture 1 overlaps with posture 3 which has closer mean value, but the structure of posture 1 is different from posture 3. We note that this method (PSI) preserves the inherent structure of the data. PSI index is not only a mere quantification of overlap of two postures in 2D space; rather it compares the similarity between two postures in the multidimensional synergy space. The advantage of using synergy space is that the reproduction of joint angles is relatively accurate. A significant value of PSI index between two postures shows that there is a greater overlap of two postures in synergy space. If the value of PSI is smaller between two postures, it means that the overlap of two postures in synergy space is lesser.

In this example of synthetic data, the value of PSI between postures 1 and 2 is 0.0757, whereas PSI between postures 1 and 3 is 0.0587. In this case postures 1 and 2 are more similar in comparison to postures 1 and 3. Structure and variance (SD) of posture 3 are different from that of postures 1 and 2. Hence we need more synergies for transforming from posture 1 to 3.

We performed normalization of PSI matrix using Eq. (6).

$$\text{Normalised } PSI_{i,j} = \frac{PSI_{i,j}}{\max\left(PSI_j\right)}. \tag{6}$$

Normalized PSI index gives the values for postures 1, 2, and 3 as listed in Table S3.
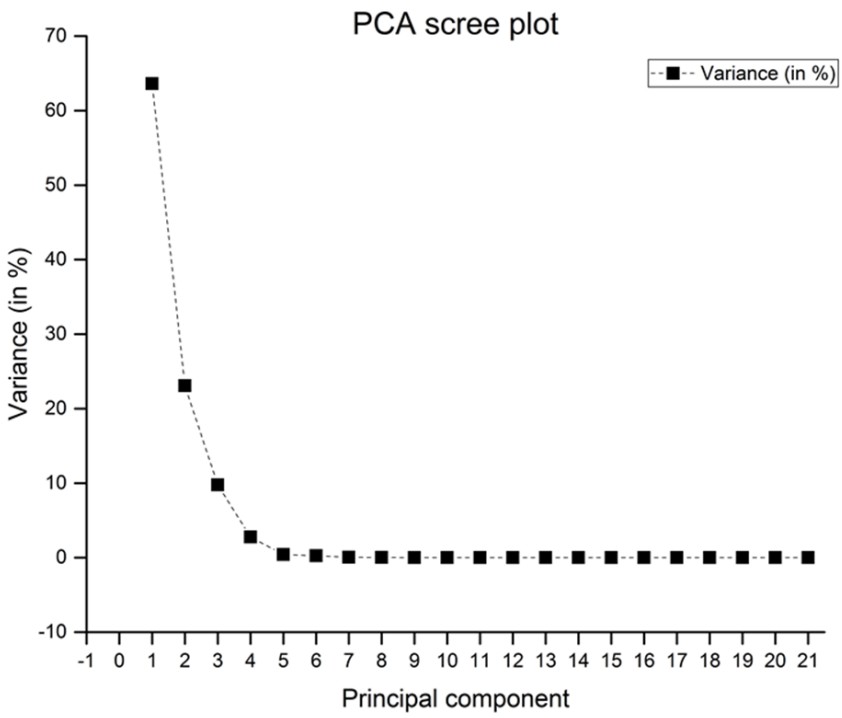

**Figure 4** **Scree plot for actual postures with three trials.** PC1 explains approximately 63.6% of variance, PC2 and PC3 explains nearly 23% and 9% of variance respectively.

## PSI for the experimental test dataset

We computed the posture similarity index (PSI) for real experimental data. Each posture contained seventy samples and 21-dimensional data. For the purpose of illustration, we randomly selected a single posture with three different trials. We normalized the data before transforming from joint angle space to synergy space using PCA. The first synergy component explains nearly 63% of the variance. The second synergy component explains roughly 23% of total variance and the third synergy component explains around 9% of the variance. We have presented the screen plot in Fig. 4.

Kinematic synergies generated by PCA by projecting joint angles on synergy space will give 21 different synergy components.

We considered all the 21 PCs for further analysis. We used the projection matrix $W$ ($21 \times 21$) for transforming from joint angle space to synergy space. We computed the covariance matrix ($\Sigma_i$ for $i = 1:21, 21 \times 21$) for individual postures in synergy space. Diagonal elements of covariance matrix explain variance in each PC direction. For generating PSI index, we multiplied individual PC ratio ($F$) with respective weights ($\lambda$) and then summed up. PSI indices are presented in Table S4. Table S5 presents the normalized PSI.

The normalized PSI index shows that postures are highly similar to each other in synergy space. This is expected, as we have performed analysis for a single posture with different trials. However, Note that, even for the same posture across different trials, PSI is not

expected to reach the theoretical maximum of 1. This is because, although functionally it is the same posture, the exact joint angles may vary and hence PSI may not reach the theoretical maximum. However, we expect the PSI to be higher across trials of the same posture when compared to PSI across postures.

This example shows that our method works well both on real experimental and synthetic data in synergy space. Finding PSI for the set of hand postures will give a quantification measure from the perspective of overlap in synergy space.

### Representative postures

For finding representative postures, we further used PSI for identifying the most significant set of postures ("Eigen postures"). We added column-wise PSIs—this represents the overall similarity that a posture has with all other postures. We then added row-wise to determine the total PSI value. Using these values, we developed a relative PSI index presented in the following Eq. (7).

$$\text{Relative } PSI_c = \frac{\sum_{i=1}^{100} PSI_{c,i}}{\sum_{i,j=1}^{100} PSI_{i,j}}. \tag{7}$$

In Eq. (7), $C$ is column-wise PSI value.

Further, we sorted the relative PSI in ascending order. The leftmost element represents the posture with the lowest similarity and as we move from left to right relative PSI index increases. The rightmost element has maximum relative PSI. Cumulative relative PSIs were computed by summing up the sorted relative PSI curve. We computed the first derivative of cumulative relative PSI and determined its peak. Postures on the right-hand side of the peak value are representative postures. We call these postures as representative postures, since they help in deriving other postures based on kinematic synergies. The representative postures are more associated with many other postures from the synergy point of view.

The peak of the derivative of relative PSI is not expected to happen at the same posture number across participants. Also, it is not necessary for the same postures to lie on the right side of this peak. In other words, it is not theoretically necessary for the same postures to be representative in all participants. The most common postures found among representative postures in different participants are presented in the Results section.

## RESULTS

### Kinematic synergy using linear method

As shown in Fig. 5, DIP vs PIP joint angle shows that there is a strong linear relationship ($r^2 = 0.997$) between the joint angles of DIP and PIP. These results are in line with previous studies (*Cerveri et al., 2007*; *Cobos et al., 2008*; *Cobos et al., 2010*; *Bullock, Borràs & Dollar, 2012*).

We performed PCA by centering data and using Singular Value Decomposition (SVD) algorithm. Results were averaged across participants and presented in Fig. 6. From Fig. 6 we observe that in synergy space the 1st PC explains approximately 40% of total variance and the nearly first five PCs explained 84% of total variance in data. The first 10 PCs explain approximately 95% of the total variance.

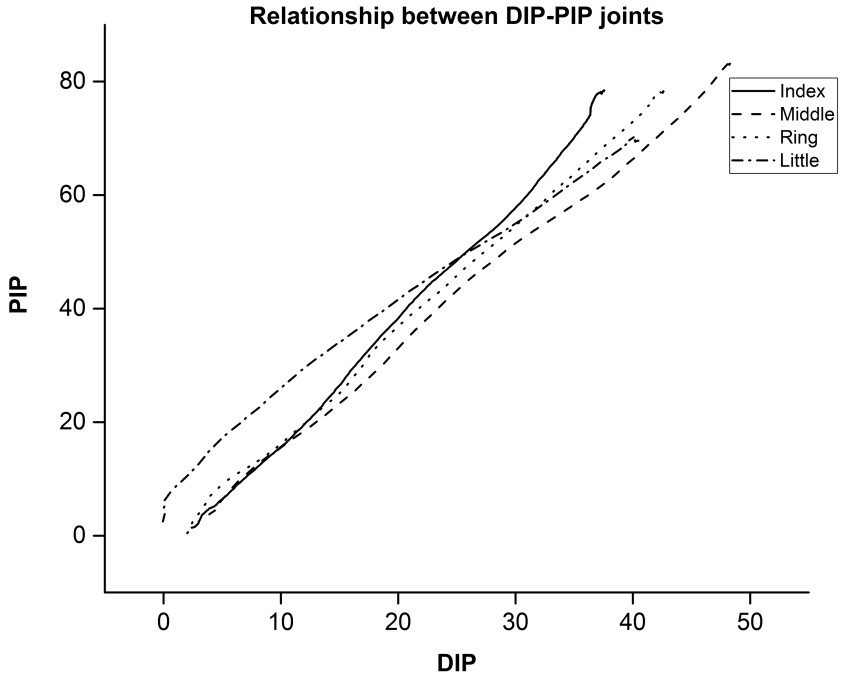

**Figure 5  Mean value between DIP-PIP joints for a single trial across participants.** Joint angle relation between distal interphalangeal (DIP) and Proximal interphalangeal (PIP) joints.

Static joint angles were reconstructed using 1st N PCs (i.e., Synergies) where N varies from 2 to 21. As expected, joint angle reconstruction error reduces as more number of kinematic synergies are involved. Joint angle reconstruction error was computed across all joints, across twelve participants as shown in Fig. 7. The shaded region represents standard error of mean across participants.

From the result in Fig. 7, we find that the error in joint angle reconstruction is higher when considering few synergy components. The experiment involves not only activities of daily living but also ASL and aesthetic (Bharatanatyam) postures.

The *posture similarity index* analysis was run for all participants separately. For analysis, one out of five trials was selected randomly for all the 100 postures. PSI matrix ($100 \times 100$) gives the similarity between any two postures in the dataset. We have presented the PSI matrix for a single participant in Fig. 8.

Based on the PSI matrix shown in Fig. 8, posture 11 was selected and compared with the rest of the postures for illustration purpose. In Fig. 9, we have presented postures in the descending order of posture similarity index for top five PSI values.

Based on PSI value, posture 15 is most similar to posture 11. Postures 37, 58 and 3 have lower similarity with posture 11. Posture 11 and 15 share common features such as the posture of the thumb. However, the postures of other fingers are different. The high similarity between postures 11 and 15 means that there is more significant overlap between these postures. Posture 37 which shares a similar thumb position as posture 11 has large PSI value. In posture 58, index and middle fingers are straight, similar to posture 11, which

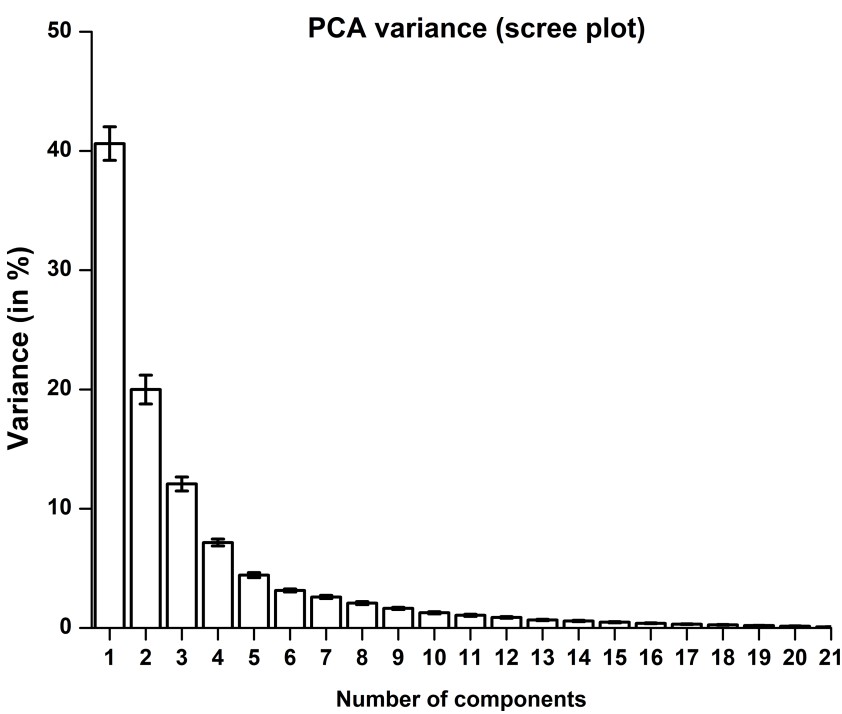

**Figure 6** **PCA variance explained by a principal component.** The 1st PC explains nearly 40% of variance and the 2nd PC explains nearly 20% of variance. Error bar represents standard error of mean across participants.

causes more substantial similarity; but ring and little fingers are flexed. Posture 3 shares common features such as the abduction of middle, ring, and little finger which results in high PSI value. For each participant, the PSI matrix is unique.

## Representative postures

For finding representative postures, relative PSI values were computed by summing all the PSI values for a posture and finding the ratio with total PSI value of a matrix for each posture separately, as shown in Fig. 10A. Postures were rearranged in ascending order based on relative PSI index and the first derivative of relative PSI was determined as shown in Fig. 10B.

The peak value was measured for each participant. For comparison, we used the smallest peak across participants as the peak. Since the representative postures are not expected to be the same across participants, we present the most common postures in Fig. 11 (We have presented only those postures found across at least seven participants).

## Statistical analysis

Many representative postures in the dataset included more "special" postures rather than common postures. However, the number of "special" postures used in the experiment was comparatively more than common postures. To rule out the possibility that the representative postures had more "special" postures purely because they were larger in number in the overall dataset, we normalized the number of representative postures by

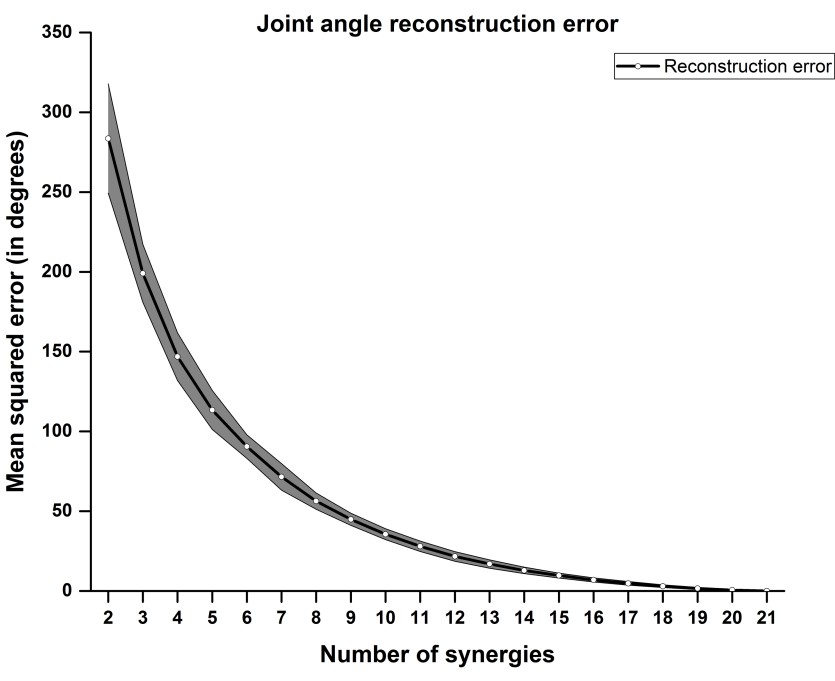

**Figure 7 Joint angle reconstruction error.** Mean squared error across postures were plotted against number of synergy components counted. Joint angle reconstruction error reduces as more number of synergy components were introduced.

the respective number of postures present in the actual dataset separately for "special" and common postures. The paired $t$-test showed that the number of common postures is significantly lesser than the number of special postures in a set of representative postures. Further, we performed one-way repeated measures ANOVA, with the type of posture (i.e., common Vs special) as a factor on the number of postures present in the representative postures. We found that the number of special postures is significantly higher ($F_{(1,11)} = 39.99$; $p < 0.001$) than the number of common postures (Mean ± standard error, common Vs "special": $16.91 \pm 0.96$ Vs $29.75 \pm 0.98$). These findings revealed that for the given dataset more number of "special" postures are represented in synergy subspace in comparison with common postures. We illustrate these results in Fig. 12.

## DISCUSSION

This article showcases a novel approach for comparing hand postures using synergies (PCA). Our method uses a ratio of variances in Eigen directions generated by PCA for comparison of two static hand postures. Posture similarity index (PSI) is a measure of similarity of two postures in the synergy space. Further, we used this index to identify representative postures from a large pool of various postures including common postures (activities of daily living ADL) and "special" postures. Our approach which involved combining the original structure of hand postures allows the relative comparison of the overlap in synergy space between two postures. We use this index to identify a set of

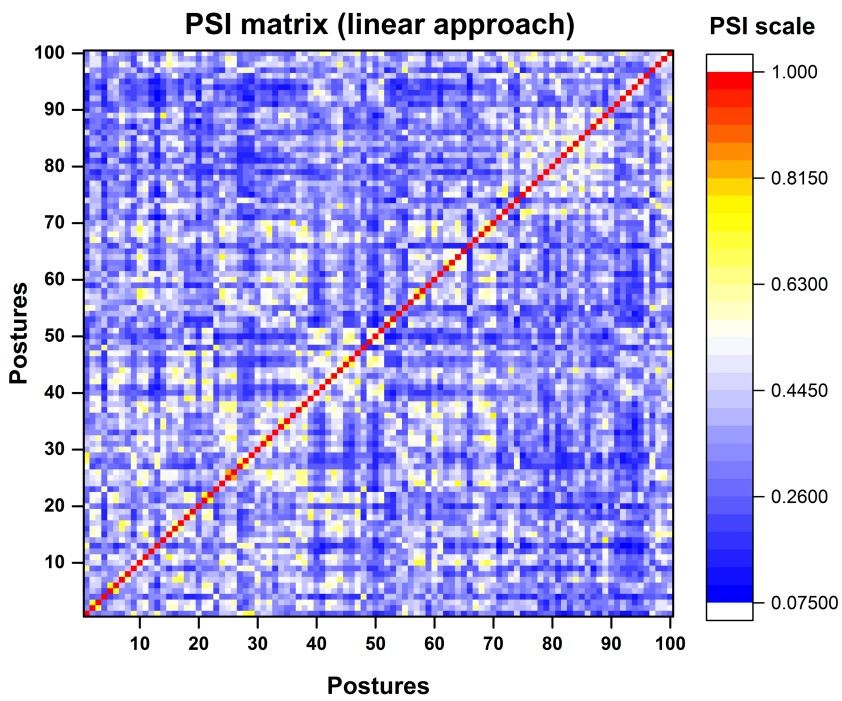

**Figure 8** **Colour map representation of PSI matrix.** Dark red colour represents very high similarity between postures in synergy space where as blue colour represents low similarity between postures.

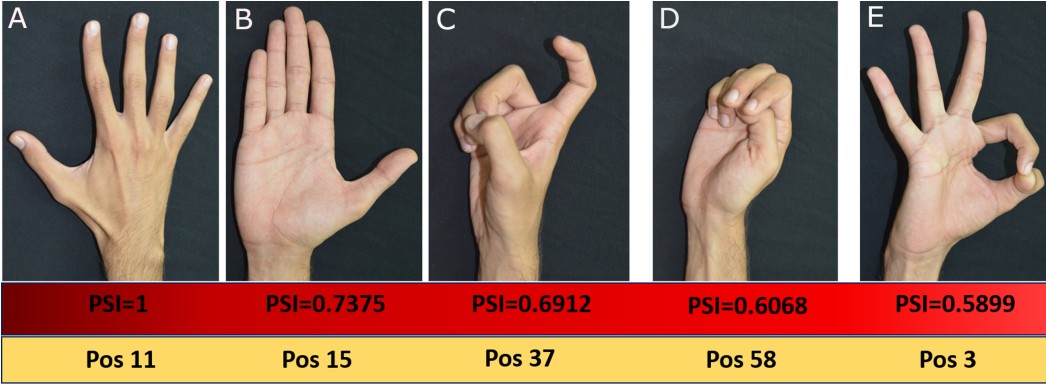

**Figure 9** **Top five Postures based on PSI values.** All photographs by Varadhan SKM. Hand model: Nayan Bhatt.

representative postures. We discuss the implications of our findings in the following sections.

## Kinematic synergies during performance of a dynamic task

In the literature, several studies have documented evidence for synergies in constrained and unconstrained exploration tasks. These have involved environmental constraint exploration, unconstrained exploration, and grasping of real and imaginary objects

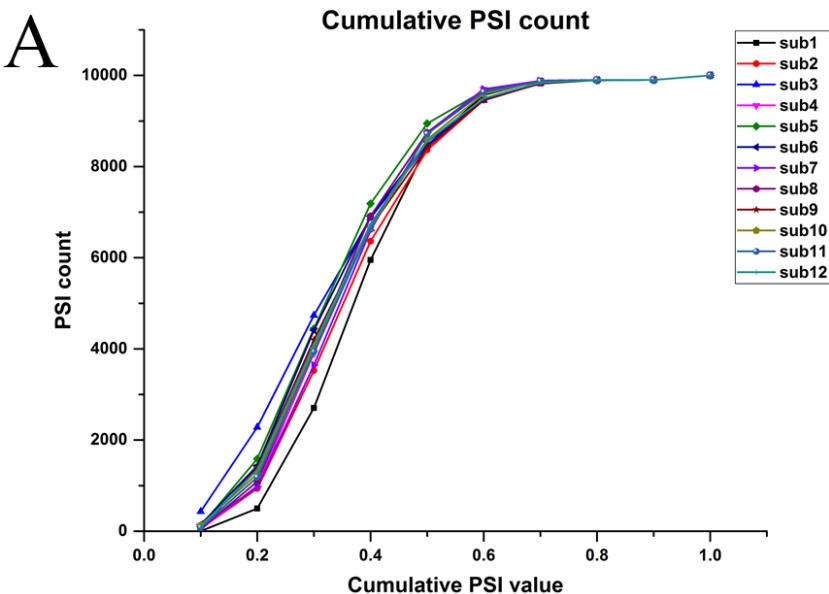

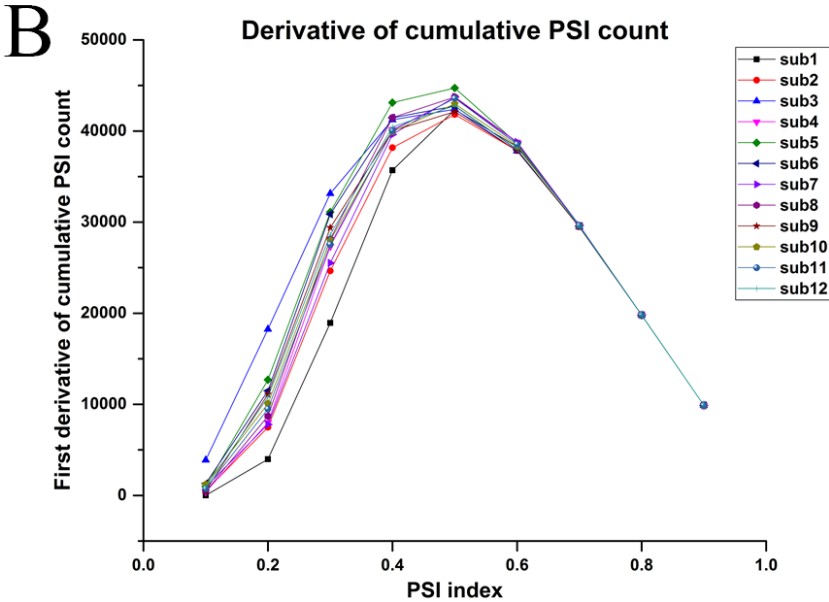

**Figure 10** (A) Cumulative relative PSI index. (B) First derivative of relative PSI index for all participants.

(*Santello, Flanders & Soechting, 1998*). *Thakur, Bastian & Hsiao (2008)* documented the existence of task-independent synergy components across several tasks, while the participants were blindfolded and were allowed relatively free haptic exploration of several real-life objects. They also found that some of these synergies are similar across several participants. In more recent studies, (*Della Santina et al., 2017*; *Eppner et al., 2015*)

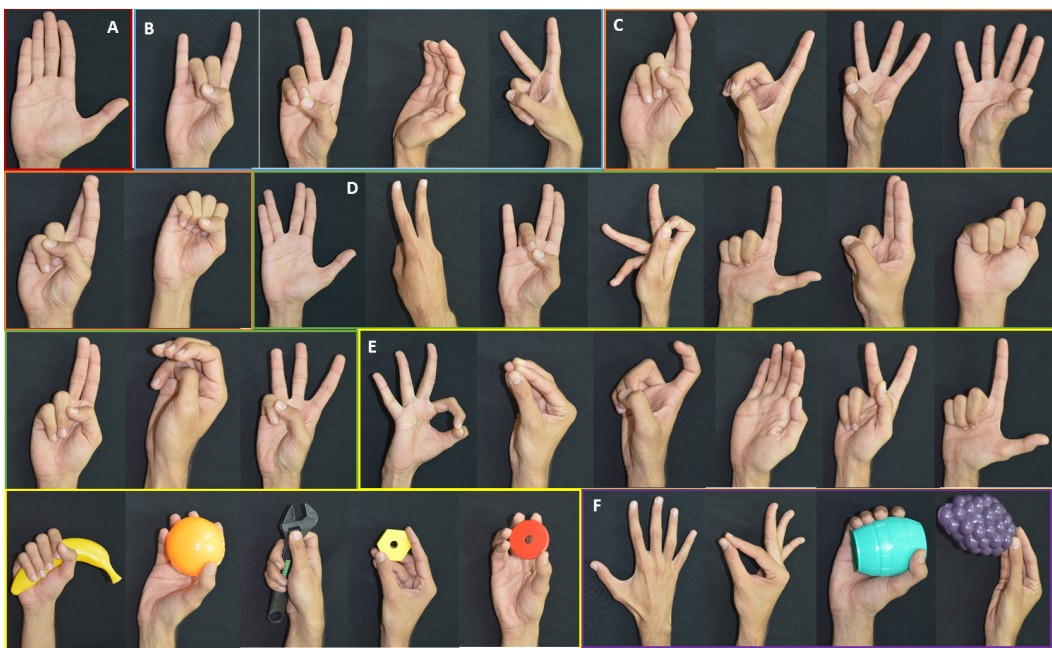

**Figure 11  Set of representative postures present in most participants.** (A) Most common representative posture—present in all 12 participants. (B) Common postures observed in 11 participants. (C) Common postures observed in 10 participants. (D) Common postures observed in nine participants. (E) Common postures observed in eight participants. (F) Common postures observed in seven participants.

constrained exploration has been used to document the existence of kinematic synergies. In our study, we have used one hundred different postures, comprising of relatively new ("special postures", including ASL, Bharatanatyam postures) and familiar ("common", daily life postures) postures in addition to some real-life object grasping postures. Our results from the synergy perspective are in-line with previous studies (*Santello, Flanders & Soechting, 1998*). However, it must be noted that in our study we have analyzed static posture (70 data samples per trial per posture), but not the actual movement (kinematics involving exploration, pre-shaping). Our goal here was to develop an index that captured the similarity of postures in the synergy space. Hence, we considered each posture and compared with other postures within the synergy (PCA) space. Hence, our "representative postures" are "representative" among the 100 postures considered, but not a combination of several postures as in synergies. Since we used a relatively large number of postures, we believe that our representative postures could also be "representative" in a real-life sense, although this is just a speculation. This is because the human hand can perform a myriad number of postures and we have only considered 100 of those postures.

## Use of human hand synergies in the development of robotic hands

One application of human hand synergies involves the development of dexterous robots and prosthetic devices. Several groups have developed and documented synergy based robotic hands (*Ciocarlie & Allen, 2009*; *Brown & Asada, 2007*; *Fani et al., 2016*). It is unclear, however, whether anthropomorphism is the best possible strategy for using

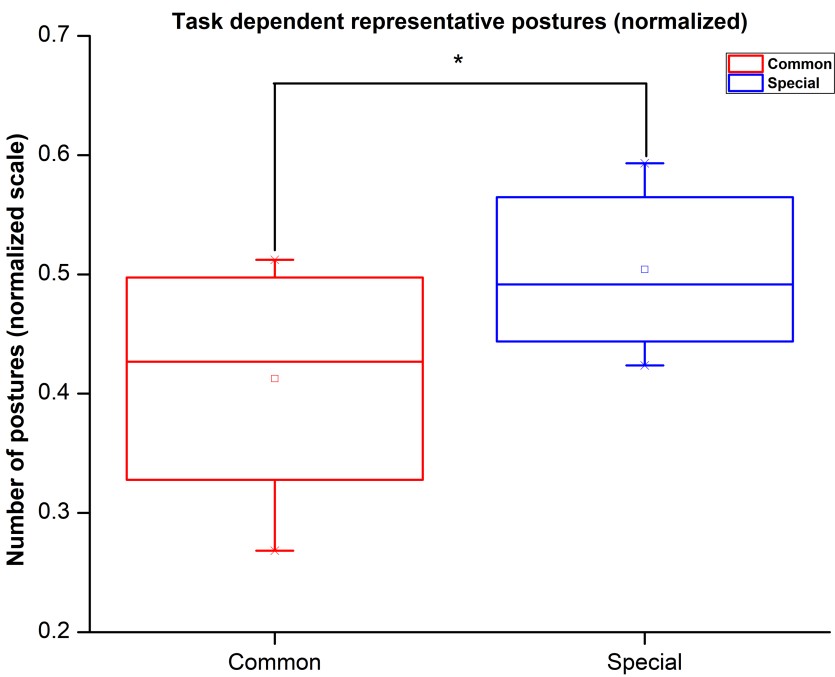

**Figure 12** **Task dependent representative posture count.** Number of representative special postures are significantly large in synergy space in comparison with common postures.

synergies in building better robotic hands (*Santello et al., 2016*). However, in cases involving an interaction of a human with robots, anthropomorphism is arguably a desirable feature. It has also been suggested that a mechanism similar to motor synergies in hand may underlie the function of mirror neurons (*D'Ausilio, Bartoli & Maffongelli, 2015*). Anthropomorphism index (AI) has been used for comparing human hand posture with artificial hand using a projection of multi-dimensional posture into two-dimensional nonlinear subspace (*Feix et al., 2013*). The method uses a comparison on a cell to cell basis in two-dimensional space. However, this method assumes that we can reconstruct the actual posture from two-dimensional nonlinear subspace. In our study, we developed the PSI, which compares two postures performed by humans in the synergy space. Here we project the postures to the synergy space to perform the comparison. PSI considers the variance in all directions which assure accurate hand posture reconstruction. A significant value of PSI represents high similarity between two postures and gives an indication of more substantial overlap in synergy space between these postures. PSI values are not always intuitive due to the abundance in the motor system.

**Representative postures and synergies—Possible neural mechanisms**
Synergies are neuronal structural units which the central nervous system uses as the basis for generating any new posture (*Santello, Flanders & Soechting, 1998*; *Mason, Gomez & Ebner, 2001*; *Leo et al., 2016*). In our study, we use representative postures. These are individual elements that are most represented through multiple postural synergies, not elements of synergy. One recent study has suggested that a small number of muscle synergies may

underlie a large number of grasps (*Scano et al., 2018*). Our results, in the joint angle space, agrees with these results; further, we present the common postures across participants. Several studies have documented evidence for synergistic movement of fingers through neural stimulating (*Gentner & Classen, 2006*; *Gentner et al., 2010*) in primary motor cortex, similar to a study that involves grasping of an imagined object (*Santello, Flanders & Soechting, 1998*; *Leo et al., 2016*). These findings provide strong support to the notion of a synergistic representation in brain areas. We believe that our representative postures are those that seek equal action manifold in comparison to our large set of postures. Our finding also suggests that significantly large number of "special" postures are part of representative postures rather than common postures. We believe that this may be because a smaller number of synergies can capture overlearned tasks. But such smaller number of synergies may not be able to capture the tremendous behavioral flexibility in the human system (*Gentner et al., 2010*). Hence, common postures occupy lesser space and are more concentrated whereas "special" postures are spread across synergy space. Our method captures this effect reasonably well.

## Concluding comments

In this study, we have defined, developed and demonstrated the use of a synergy based posture comparison index, PSI. It is also useful to classify postures as "representative" or otherwise. A limitation of the current approach is that comparison between two hand postures depends on the complexity of hand model. Our hand model consists of 21 DoF model, which does not account for the palm arch and wrist joints. PSI may give more insightful results for a complex model involving multiple degrees of freedom. Distribution of data may provide more hidden information about hand postures and will be able to produce more accurate results.

### Funding

This work was funded by the Department of Science and Technology of India's Cognitive Science research initiative (grant number: SR/CSRI/97/2014(G)). The funders had no role in study design, data collection and analysis, decision to publish, or preparation of the manuscript.

### Grant Disclosures

The following grant information was disclosed by the authors:
Department of Science and Technology, India's Cognitive Science research initiative: SR/CSRI/97/2014(G).

### Competing Interests

The authors declare there are no competing interests.

## Author Contributions

- Nayan Bhatt conceived and designed the experiments, performed the experiments, analyzed the data, prepared figures and/or tables, authored or reviewed drafts of the paper, approved the final draft.
- Varadhan SKM conceived and designed the experiments, contributed reagents/materials/analysis tools, authored or reviewed drafts of the paper, approved the final draft, acquired funding for the study.

## Human Ethics

The following information was supplied relating to ethical approvals (i.e., approving body and any reference numbers):

Experimental procedures were approved by the Institutional Ethics Committee of IIT Madras (Approval number: IEC/2016/02/VSK-1/11).

## Data Availability

Bhatt, Nayan; SKM, Varadhan (2018): Posture similarity index: a method to compare hand postures in synergy space. figshare. Dataset. https://doi.org/10.6084/m9.figshare.7325495.v1.

## Supplemental Information

Supplemental information for this article can be found online at http://dx.doi.org/10.7717/peerj.6078#supplemental-information.

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
