# Peer review of "Posture similarity index: a method to compare hand postures in synergy space"

_PeerJ, doi:10.7717/peerj.6078_

## Round 0.1 · original submission · Major Revisions

As you will find, while the reviewers assessed your work to be of interest, they raised a number of substantial concerns about the manuscript. In particular, the reviewers indicated: (a) the need for improving the clarity of the manuscript; b) the need for greater clarity regarding the novelty of the study; (c) aspects of the Discussion's content. Again, these and the reviewers' other major and more minor comments can be found below.

Reviewer 1 ·

Basic reporting

In all the manuscript, the English language must be improved. The current phrasing makes comprehension difficult. Some examples: lines 28-29, 60-64, 83-86, 135-136.

Abstract

Line 27: “Wonderful” may not be the most descriptive term while “several” may be a bit limited.
In “background” (Lines 26 to 33), the link between the idea of “effort” to adopt/change hand posture, and the goal of the study to quantify the similarity between hand postures is unclear.

Line 41, PCA is a widely used dimensionality reduction technique, already successfully applied on hand postures characterization (e.g. Gentner & Classen, 2006; Koul, Cavallo, Ansuini, & Becchio, 2016). What is the novelty of the present study? Authors should stress the added value of their study.

Introduction

Line 61: This paragraph is very difficult to understand. It starts with “however” but to me the presence of redundancy is equivalent to the sentences above, explaining the necessity to coordinate many muscles and articulatory joints.

Line 68-69: “…execution, sometimes even managing every single variable for executing a task successfully”, the authors should reference this hypothesis.

Line 70-71: “…the CNS forms a synergy (a group of relatively independent variables) (Latash, 2008b), that combines multiple elements which result in the successful performance of the task”. This is a very unusual and obscure definition of synergies. What are “relatively independent variables”? What are the “multiple elements” combined? And what is the link between these synergies, these variables and these elements?

Knowing that SNC control muscles or group of muscles (that results into kinematic trajectories), how invariances at the kinematic level are reliably representing the control command?
The authors should also define the term “eigenpostures” (e.g. lines 83-86).

At the end of the introduction, the link between effort, synergies and posture similarity index is still unclear.

In the rest, they justify this study by mentioning the necessity to increase the degrees of freedom and define from that, a similarity index. It not really clear to me the innovation here and in the authors should clearly argue better for their technique in the introduction.

Experimental design

Method & Results
Lines 180: “Cut-off frequency for the filter was set to 4.5Hz for removing any possible effect of physiological tremors in static postures”. This cut-off value is surprising and low, the authors should reference studies using a similar value.

Lines 182: “Trials were concatenated”. Is the concatenation of different trials creating artifact (e.g jump)? Did the authors control this?

Lines 214: “By projecting hand postures in synergy subspace, we can take the dimensionality reduction approach (Santello, 1998) or take the stance that the control signal that helps controlling multiple degrees of freedom simultaneously (Todorov, 2002; Todorov et al., 2004).”
Among many others, this formulation is very difficult to understand. In all cases, Todorov’s model is not against the idea of a dimensionality reduction approach, it’s rather a complementary account.

Lines 222-224: “Assuming there is no relation between trials we selected a random trial from five trials for every posture separately, resulting in 100 selected trials per subject (one trial per posture).” Why did they proceed in this way?

Some of the figures may need some additional work to improve quality (i.e. F2), some of them are redundant or not necessary (i.e. F2; F8 panel B) or can be merged in one figure (i.e. F1 and F3; F4 and F5). Authors may want to move some of the figures or tables in the supp. materials (i.e. Table)

Validity of the findings

Discussion
Line 428: “This article showcases a novel approach for comparing hand postures using synergies (PCA).” Unfortunately, it is not clearly visible why this method should be new, see my comments in introduction.

Line 435: The link between “kinematic effort” and similarities between postures is still unjustified.

At the end of the discussion, when the authors go through the goal of the study, I mainly see the description of a method to extract kinematic synergies and compare between them. If this is the goal, they should better justify how such a commonly used technique (PCA) is here used in a “different” way.

·

Basic reporting

-The work targeting building similarity index between for hand postures. From a total of 100 different postures, individual finger Kinematics were captured by wearable tracking sensor system. PCA was used to extract kinematic synergies. Experimentally, the developed PSI was validated to identify various postures.
-The work from technical point of view is interesting, however, the usability of the work and the future implementation is still weak and not very much connected with the result. I believe this part (the connection) should be much enhanced.
-The introduction is kind of long and discusses various issues that are not in the essential point of the work direction. I believe the introduction could be enhanced/shortened.
- References are old

Experimental design

well covered

Validity of the findings

Not as I expected. Some figures of the finding are hard to understand. What was the original postures represented on the screen? All individuals were requested to follow the same postures?

Additional comments

- The idea of the work is interesting and the reported resulting are promising. The above issues are needed to make the story worth to read.

---

## Round 0.2 · Minor Revisions

Generally, I feel that the quality of the manuscript has improved considerably, Only minor changes are required from Reviewer 1. Please consider seeking the help of a proofreader with good written English skills.

Reviewer 1 ·

Basic reporting

The writing’s quality improved but a lot of sentences are still a bit confused, I will suggest asking a native English speaker to correct the manuscript before publication.

Introduction:
Page 3, Line 69: “low” instead of “lower”, there is no comparison here so ne need for “-er”
Page 3, Line 71: “small” instead of “smaller”
Page 3, Line 76: “This has been attempted probably because of the notion that designing better grasping robots must essentially involve learning from human grasping.” This sentence is complicated and should be reformulated
Page, Line 91: “field” instead of “fields”

Materials and Mehods
Figure 1: increase the size of the font in A’s labels
Figure 2: explain in the caption, why three pictures on the screen? Why one is colored? And why is there a F letter on the right?
Page 7, Line 214: “The experiment consisted of two different tasks, the first task was performing internally constrained postures which include 70 postures, and the rest 30 are constrained postures which involve object manipulation.” This sentence should be reformulated.
Page 8, Line 228: “All the trials were carefully observed by the experimenter and any trial in which the posture was not faithfully performed was repeated.” How many trials on average were repeated?
Page 8, Line 230: “All joint angles” instead of “All the joint angles”
Page 9, Line 253: Suppress “there is”
Page 10, Line 283: “slight” and “slightly” are a bit redundant, please find a synonym
Page 14, Line 395: “Individual PC ratio (F) is multiplied by respective weights (λ) and summed for generating PSI index as in following are present in Supplemental Table S4.” This sentence should be reformulated.
Page 14, Line 408: ”So, in this case, since we chose the same posture (but different trials), so that the data has a relatively large overlap.. changing from one trial’s posture to another requires minimal effort.” This sentence should be reformulated.
Page 14, Line 412: “An advantage of this method is it compares and quantify similarity in synergy space rather than joint angle space.” This sentence should be reformulated.

Results:
Page 15, Line 442: “with previous studies” instead of “with the previous study”
Page 16, Line 456: “one of the possible reasons is task involved in the experiment” incorrect wording
Page 17, Line 489: “only those postures found in 7 subjects are more are presented here for illustration purpose and brevity” incorrect wording
Page 17, Line 492: “include” instead of “includes”
Page 17, Line 493: “the number […] was” instead of “were”
Page 17, Line 503: Specify associated statistical results
Page 17, Line 504: “in in”

Discussion:
Page 18, Line 509: “exploits use” incorrect wording
Page 18, Line 523: “we used idea further” incorrect wording
Page 19, Line 559: “take into account of” suppress the “of”
Page 19, Line 562: “Sometime PSI does not result in intuitive numbers while comparing two postures, possible major reason behind this because of degrees of freedom and the large number of tasks involved.” Incorrect wording

Experimental design

satisfactory

Validity of the findings

satisfactory

Additional comments

Authors have well clarified the manuscript and made clear the goal of the study: creating an index to compare posture. As explicitly stated now, this manuscript describes a new method.

However, theoretical insights for neuroscience or robotic are for the moment absent.

·

Basic reporting

The revised manuscript is well enhanced based on the previous comments. Figures are well designed now and the story is clear.

Experimental design

Well defined

Validity of the findings

Enough and represent the target of the work

---

## Round 0.3 · accepted · Accept

Congratulations on your latest contribution.